# Research Trends in Motivation and Weight Loss: A Bibliometric-Based Review

**DOI:** 10.3390/healthcare11233086

**Published:** 2023-12-01

**Authors:** Uroš Železnik, Peter Kokol, Jasmina Starc, Danica Železnik, Jernej Završnik, Helena Blažun Vošner

**Affiliations:** 1Faculty of Health and Social Sciences Slovenj Gradec, 2380 Slovenj Gradec, Slovenia; dekanica@fzsv.si (D.Ž.); helena.blazun@zd-mb.si (H.B.V.); 2Health Education Center, Community Healthcare Center Ptuj, 2250 Ptuj, Slovenia; 3Laboratory for System Design, Faculty of Electrical Engineering and Computer Science, University of Maribor, 2000 Maribor, Slovenia; peter.kokol@um.si; 4Faculty of Medicine, University of Maribor, 2000 Maribor, Slovenia; 5Faculty of Business and Management Sciences, University of Novo Mesto, 8000 Novo Mesto, Slovenia; jasmina.starc@uni-nm.si; 6Community Healthcare Center Dr. Adolf Drolc Maribor, 2000 Maribor, Slovenia; jernej.zavrsnik@zd-mb.si

**Keywords:** obesity, weight loss, motivation, synthetic knowledge synthesis, bibliometrics, content analysis

## Abstract

Obesity is a complex disease that, like COVID-19, has reached pandemic proportions. Consequently, it has become a rapidly growing scientific field, represented by an extensive body of research publications. Therefore, the aim of this study was to present the research trends in the scientific literature on motivation and weight loss. Because traditional knowledge synthesis approaches are not appropriate for analyzing large corpora of research evidence, we utilized a novel knowledge synthesis approach called synthetic knowledge synthesis (SKS) to generate new holistic insights into obesity research focusing on motivation. SKS is a triangulation of bibliometric analysis, bibliometric mapping, and content analysis. Using it, we analyzed the corpus of publications retrieved from the Scopus database, using the search string TITLE-ABS-KEY((obesity or overweight) and “weight loss” and motiv*) in titles, keywords, and abstracts, without any additional inclusion or exclusion criteria. The search resulted in a corpus of 2301 publications. The United States of America, the United Kingdom, and Australia were the most productive countries. Four themes emerged, namely, weight loss and weight-loss maintenance through motivational interventions, lifestyle changes supported by smart ICT, maintaining sustainable weight with a healthier lifestyle, and weight management on the level of primary healthcare and bariatric surgery. Further, we established that the volume of research literature is growing, as is the scope of the research. However, we observed a regional concentration of research and its funding in developed countries and almost nonexistent research cooperation between developed and less-developed countries.

## 1. Introduction

Numerous factors, such as accelerated rates of scientific knowledge doubling, web/internet-based methods of scholarly communication, exponential growth in scientific information production, increasingly faster cycles of technological innovations, and the open access and open science movements, have significantly increased the complexity of research evidence synthesis. Simultaneously, these phenomena have led to a surge in the availability of research literature in machine-readable formats [1]. As a result, these developments have not only presented challenges but also provided opportunities. A vast amount of scientific evidence is now available in a digital format that can be digitally synthesized using computer algorithms.

Taking advantage of this opportunity, Kokol et al. [1] developed a novel synthetics knowledge synthesis methodology (SKS), based on the triangulation of the following approaches:Distant reading [2]: an approach for understanding the canons of literature not through close (i.e., manual) reading but by employing computer-based technologies such as text mining and machine learning;Bibliometric mapping [3];Content analysis [4].

In this sense, SKS overcomes some of the weaknesses of traditional knowledge synthesis approaches such as literature reviews or meta-analyses. Firstly, SKS requires fewer resources, is less time- and labor-consuming, and is conducted semi-automatically. As a result, it can be applied to a large number of large corpora of thousands or tens of thousands of publications, and not just on a small sample of manually selected publications as in other approaches. The results can be reproduced and the research field can be explored in a holistic way with less personal bias.

This triangulation integrates quantitative and qualitative knowledge synthesis as an augmentation of the traditional bibliometric analysis of publication metadata with machine-learning-supported insight into the patterns, structure, and content of scientific evidence [5].

Obesity is a complex disease that, along with COVID-19, has reached pandemic proportions [6]. At the same time, it poses a significant long-term health problem, with enhanced involvement of younger populations. Globally, there are more people who are obese than underweight and also more deaths are connected to obesity than underweight, with the exception of some parts of sub-Saharan Africa and Asia. Overweight and obesity and related chronic diseases (high blood pressure, diabetes, cardiovascular diseases, etc.) are largely preventable by the consumption of healthier foods and regular physical activity, with these being the easiest choices and also the most accessible, available, and economically affordable [7].

Nevertheless, the promotion of and access to healthy lifestyles are needed to consider the full effect of individual responsibility. Therefore, at the societal level it is important to support individuals through the sustained implementation of evidence-based and population-based policies in concordance with recommendations to prevent obesity. An example of such a policy is a tax on sugar-sweetened beverages. Moreover, low-income individuals must gain access to more physical activity options and healthier dietary choices that are economically affordable for everyone [7].

With over half a million articles, including more than 50,000 focused on weight loss, it is also a rapidly growing scientific field with an extensive body of research publications. As stated above, the traditional approach to knowledge synthesis is insufficient for obtaining a holistic and comprehensive understanding of the obesity research landscape. While various conventional bibliometric studies concerning obesity and being overweight have been performed [8,9,10], none have focused on motivation or have been qualitatively oriented. To address this gap, we utilized the SKS methodology to generate new insight by answering the following research questions:To what extent are the volume and scope of the research sufficient in terms of importance?How is the research dispersed among countries (focusing on developed and less-developed countries) to support the World Health Organization’s global health initiatives and optimal country cooperation and knowledge transfer?What are the more prolific information titles/journals that inform the scientific community about research in the field of motivation in obesity, and what are the sources through which the authors have the greatest opportunity to inform the community about the results of their research?Which founding bodies are more prolific in sponsoring the research?What are the most prolific research themes?

The choice to focus on motivation research in obesity and weight loss was motivated by the belief that motivation is a crucial factor in weight loss for a myriad of stakeholders, such as health professionals, social workers, and obese citizens. This study can help them gain new insight into the topic, deepen their knowledge, or inform them about the trends and essential themes in obesity and motivation research. To the best of our knowledge, no similar quantitative and qualitative bibliometric study that provides a holistic overview of the current state from its beginnings to the present has been conducted so far.

Sandoval et al. [11] describe intermittent fasting in overweight individuals as beneficial for improving body composition by reducing fat mass and preserving muscle mass. They also state that additional studies are needed in the future to better explain the effect of intermittent fasting on body composition.

## 2. Methods

The primary advantage of bibliometrics [12,13,14] is that it supports the analysis of a vast number of publications at the macroscopic and microscopic levels and is domain independent [15]. Bibliometric mapping, a subdiscipline of bibliometrics, uses text mining, co-word analysis, and clustering algorithms to analyze the relationships and associations between pairs of bibliometric units (words, phrases, authors, countries, etc.) and visualizes them in the form of bibliometric maps. A bibliometric map is a network of nodes, where nodes represent bibliometric units; links represent relations; the proximity of nodes, their similarity; and the node size is the unit popularity. Clusters of nodes represent strongly associated units. One of the most widely used bibliometric mapping software tools is VOSViewer (Leiden University, Leiden, The Netherlands) [16].

Alternatively, content analysis is a versatile approach used in both quantitative and qualitative research. It allows for systematic and objective descriptions of phenomena and can be applied to various types of documents—in our case, research publications. Concept analysis can be used to create concepts, categories, and themes [4]. SKS is performed using the following steps:Harvest research publications on the topic of interest from the selected bibliographic database using an appropriate search string representing the research question(s) to be answered through knowledge synthesis;Use author keywords as meaningful units of information;Perform bibliometric mapping of the authors’ keywords into a cluster bibliometric map using selected bibliometric software—in our case, VOSViewer (Leiden University, Leiden, The Netherlands) [16];Analyze the links and proximity among author keywords in individual clusters to form categories.

Scopus (Elsevier, Amsterdam, The Netherlands) was chosen as the source for the bibliographic database. To form a suitable corpus of publications to answer the above research questions, we performed a search using the following search string:

### TITLE-ABS-KEY((Obesity or Overweight) and “Weight Loss” and Motiv*)

We searched titles, keywords, and abstracts for the entire period indexed in Scopus without additional inclusion or exclusion criteria. The search was performed on 27 January 2023.

The following bibliometric maps were used in the exploratory analysis:Timeline bibliometric map to analyze the evolution of terms through time;Country co-authors citation density map to analyze the country cooperation;Two author keyword maps, one for the period from 2019 to 2020 and the other for the period from 2021 to 2022, were compared to identify future research topics [17].

The search was performed in a way to, as closely as possible, achieve the study’s objectives. In this manner, we formed the search string using the keywords focusing only on weight loss in obese or overweight people and motivation. We did not use synonyms as irrelevant publications could be inserted into the corpus. However, we used the wildcard character * and quotation marks. Thus, the subsearch string “weight loss” retrieved all publications containing the words weight loss, losing weight, weight loss, and similar. We assessed the performance of the search string with a recall metric (i.e., the fraction of relevant instances among the relevant instances). In this manner, we formed a list of 30 fundamental publications related to the aim of the study and expected to be found in the corpus and 10 prolific authors. All 10 authors and 29 publications were successfully retrieved.

Descriptive bibliometrics (most prolific, countries, institutions, journals, and funding agencies) was performed using Scopus’ built-in functions. SKS was performed using the procedure described at the beginning of this section.

## 3. Results

The search resulted in a corpus consisting of 2301 publications, including 1839 original articles, 299 review articles, 50 conference papers, 26 letters to the editor, 24 book chapters, 21 editorial contributions, 38 short articles, and 4 other types of publications (Figure 1).

### 3.1. Spatial Characteristics of the Body of Research

The first publications appeared in 1968; they addressed overeating in a psychiatric population [18] and compared university-attending and non-university-attending populations regarding obesity in young males [19]. Between 1968 and 1989, research production was modest (with a maximum of 11 articles), exhibiting a slightly positive linear trend (Figure 2). The year 2000 saw exponential growth in research productivity that lasted until 2018, when it peaked at 179 articles. Subsequently, a downward trend began, with 160 articles published in 2022. The exponential trend in the new millennium and the volume of research indicates that research on motivation for weight loss is gaining importance and popularity; however, the downward trend in recent years might be a cause for concern.

Table 1 presents the 10 most productive countries out of a total of 69, whose authors cover 90.6% of all scientific production. This highlights a regional concentration of research in more developed countries, as seven of them are members of the G20, and the Netherlands, Denmark, and Spain are among the most economically developed countries with efficient health systems. Interestingly, none of the countries with the largest percentage of obese people [20] were among the most productive countries. From the authors’ point of view, it is surprising that none of the papers were published by Slovenian authors, who ranked 71 on the above list.

The most prolific institutions, among 854, were The University of Alabama at Birmingham, USA (*n* = 39); University of Pennsylvania Perelman School of Medicine, USA (*n* = 37); Duke University, USA (*n* = 37); University of Pennsylvania, USA (*n* = 36); and The University of Sydney, Australia (*n* = 36).

The 5 most prolific funding institutions, among 417, were the National Institute of Diabetes and Digestive and Kidney Diseases, USA (*n* = 223); National Institutes of Health, USA (*n* = 171); National Heart, Lung, and Blood Institute, USA (*n* = 86); National Cancer Institute, USA (*n* = 58); and Eunice Kennedy Shriver National Institute of Child Health and Human Development, USA (*n* = 49). Notably, all of these institutions are research agencies, and none is a pharmaceutical company.

These articles were published in 693 journals. Table 2 displays the 10 most prolific journals in which approximately one-third of the articles (32.9%) were cited. With the exception of two journals, all were categorized in the first quarter (Q1) of journals looking at the SJR impact factor. The impact factor values range from 0.73 to 2.54. The H-index, another important indicator of journal influence, ranges from 209 to 49. These statistics indicate that publications in the fields of motivation and weight loss are published in well-recognized and influential journals, underscoring the importance of the topic under consideration.

### 3.2. Thematic Analysis

Thematic analysis was performed using SKS and VOSViewer software, version 1.6.19 (Leiden University, Leiden, The Netherlands). The author keyword map is shown in Figure 3, and the synthesis of the results is shown in Table 3. Four themes and ten categories were identified.

#### 3.2.1. Weight Loss and Weight-Loss Maintenance through Motivational Interventions and Lifestyle Changes Supported by Smart Information Communication Technology (ICT)

##### Lose Weight through Interventions in the Form of Motivational Interviews, Physical Activity, and Changes in Behavior and Lifestyle Supported by mHealth

A positive impact of app-based mobile interventions was identified in improving nutritional behaviors on obesity indices through behavior change management provided by apps, such as goals/planning, feedback/monitoring, shaping knowledge, and social support [21] or by improving lifestyle in general [22,23]. Health coaching through video conferencing has shown positive effects on physical activity, metabolic markers, and weight loss [24]. In contrast to the above research, a systematic review of 25 studies did not reveal a significant association between mobile app use and weight loss in middle-aged or elderly patients with prediabetes [25].

##### Health Promotion with Motivational Interviews to Encourage Physical Activity

Physical activity was widely proven to be effective in weight loss and weight management, while health promotion is of the utmost importance in encouraging physical activity. It should be taught in early childhood and upheld to old age. Therefore, a study evaluating the effects of 6-month resistance and combined aerobic-resistance physical exercising showed a significant positive effect of both types of exercises [26]. Twenty subjects who completed the training program showed improved body composition with a reduction in body mass index, adipose tissue, diastolic blood pressure, and pulse wave velocity. A similar positive effect of 6-month resistance training was observed by Shranz et al. [27].

As Fanelli et al. [26] concluded, physical exercise induces a positive effect on the cardiovascular risk profile; these positive effects also persist after a brief discontinuation, and physical exercise reduces early signs of autonomic dysfunction. Resistance training is an exercise modality in which overweight and obese adolescents can excel and which can, therefore, positively affect their psychological well-being [27].

Self-determination theory, as an explanatory framework, combined with motivational interviewing and used as part of a standard weight loss program based on physical activity practice in obese adolescents, resulted in a larger reduction in body mass index, increased autonomy, a greater rise in integrated and identified regulations, and a stronger decrease in amotivation compared to standard weight loss programs [28]. Motivational interviews also enhance adherence to obesity interventions in the pediatric population [29]. Motivation through coaching calls regarding physical activity in a safe home environment successfully reduced obesity in individuals with lower extremity amputation [30]. A systematic review of studies of weight reduction in truck drivers showed that motivational weight loss interventions, including physical activity, diet, behavioral therapy, and health promotion, may be successful [31].

##### Weight-Loss Maintenance with Telemedicine and eHealth

A systematic review of randomized controlled trials showed the effectiveness of eHealth interventions (internet, mobile apps, podcasts, emails, and similar) for the prevention and treatment of overweight and obesity in adults; however, there is insufficient evidence for weight-loss maintenance or weight regain prevention [32]. In contrast, a scoping review of studies concerned with weight loss or weight-loss maintenance supported by eHealth tools, such as activity trackers, digital scales, monitoring, goal setting, and planning through digital technology, supported the behavioral change needed to sustain a healthy lifestyle and weight loss. Similar findings were reported in another recent systematic review of patients who underwent bariatric surgery [33].

#### 3.2.2. Maintaining Sustainable Weight and Healthy Lifestyle with Digital Health Support

##### Nutrition and Weight Maintenance for Prevention of Metabolic Diseases Supported by Digital Health

It has been widely recognized that interventions, continuous evaluation, and monitoring of daily life are important for the effective treatment and management of obesity and metabolic diseases, especially if they are supported by various digital health technologies [34,35]. However, digital health interventions for weight (loss) maintenance require appropriate designs. Thus, a combination of persuasive system design principles and behavioral change techniques has been shown to be successful [36].

##### Weight Maintenance as a Lifestyle

A recent study observing active behavioral changes focusing on stress reduction showed that perceived stress is associated with eating behavior features that may weaken successful weight-loss maintenance [37]. Another study found that the effect of a lower-energy diet as a long-term lifestyle intervention was sex- and age-related. Older adults and women benefit less from it [38]. A meta-analysis and systematic review revealed that weight regain started to occur 36 weeks after the intervention; however, certain strategies may prolong that period or even prevent weight regain, such as intervention type, the presence of a dietitian on the care team, and counseling with a health professional at least once a month [39]. Healthcare resource utilization was significantly higher in weight-loss regainers than in maintainers. In addition, weight-loss maintenance was associated with a delayed onset of osteoarthritis and other obesity comorbidities. An interesting solution for weight and healthy lifestyle maintenance was proposed by Kamel and Koh [40], namely, that smart cities should support lifestyle sensing for smarter health decisions regarding obesity and overweight [41].

### 3.3. Weight Management on the Level of Primary Healthcare

#### 3.3.1. Weight Management in Overweight Patients as Part of Primary Care Supported by Qualitative Research

The US Preventive Services Task Force recommends that primary care clinicians screen all adults for obesity and refer them to an intervention called lifestyle modification, which includes diet, physical activity, and behavioral therapy. Participants lost up to 8% of their weight in 6 months and were provided with, at least, monthly counseling to prevent weight regain. They are currently introducing ICT to monitor food intake, activity, and weight to make interventions even more successful [42]. Nutritional counseling in a primary care environment is recognized as a starting point in the management of obesity [43], especially if supported by digital technology [44,45]. If integrated into family-centered pediatric weight management, it has been shown to be very successful in pediatric populations [46,47], but the sustainability of healthy lifestyle changes requires continuous support [48]. The importance of long-term counseling support has been confirmed by Kumanyika et al. [49], who showed that each additional coaching visit is associated with a 0.37 kg greater estimated 24-month weight loss.

#### 3.3.2. Barriers to Weight Loss

Weight maintenance remains a challenge, and understanding the barriers that patients experience during weight loss is crucial to improving the weight-loss and maintenance processes [50]. Factors such as a lack of resource support, logistics, knowledge regarding weight-loss interventions, understanding of the root causes of obesity, patient readiness for change, and family physicians’ perceptions about surgical weight loss have been identified as barriers to successful weight management [51]. Delahanty et al. [52] identified different types of barriers, such as a lack of self-monitoring, insufficient physical activity, internal thought/mood cues, vacation/holidays, social cues, access/weather, time management, and aches/pains during exercise. Another study highlighted habitual overconsumption; proximity and convenience of food available; momentary lack of motivation and sense of control; overeating triggers such as social media; eating with family, friends, and colleagues; provision of food by someone; emotions (e.g., sad and stressed); and physiological conditions [53].

#### 3.3.3. Bariatric Surgery

##### Bariatric Surgery and Body Self-Image

Behavior and body image can affect the relationship between obesity and weight maintenance. Body image concerns may be one reason for choosing surgery [54]. In a study conducted by Faccio et al. [55], participants reported that even one year after surgery, they behaved as if they were still obese, and greater awareness was needed to help them realize that they were no longer obese. In another study, participants experienced little or no control in relation to food and eating before the bariatric procedure, but they believed that surgery would be the first step toward gaining control in this area. One year post-operatively, they acquired established routines and had higher self-esteem. However, after two years, fear of weight gain resurfaced, and their self-image became more realistic [56]. A systematic review of post-bariatric surgery body images revealed that adapting to a new body can be challenging because of a persistently obese view of the self. Furthermore, patients are dissatisfied with excessive skin after bariatric surgery and a negative self-image is replaced by dissatisfaction [57].

##### Weight Regains after Bariatric Surgery

Post-bariatric weight gain may be associated with problematic or eating-disordered behaviors, such as loss of control of eating and depression; therefore, psychiatric treatment for such patients is advised [58]. Another important factor is patient self-efficacy expectations, which are positively associated with weight loss [59]. Positive psychological well-being, a novel approach to improving adherence by increasing positive associations with health behaviors, including physical activity after bariatric surgery, or other similar behavioral interventions, such as reward-based eating or self-compassion, could help patients maintain lifestyle changes [60,61]

#### 3.3.4. Evolution of Research

The historical trends are presented in Figure 4. Until 2012, research in the field of weight loss and motivation focused on morbid obesity, body self-image, diet, surgical techniques for weight loss, and the internet as an opportunity to motivate and search for useful information. In the 2012–2014 period, the research focus shifted to the psychological factors of obesity and weight loss in terms of eating habits and health concerns. Research on the association between obesity and diabetes was also initiated. The phenomenon of motivation in connection with behavioral change and obesity/weight loss began to be intensively researched in the 2015–2016 period. During this period, research on bariatric surgery became popular. In the 2017–2018 period, intriguing research was conducted in the fields of behavioral change interventions, motivating changes, and obstacles to weight loss. One of the more interesting concepts that appeared in this period is “self-determination theory”. Research conducted after 2018 has been devoted to the use of digital health, smart technologies, telemedicine, and social media in motivational interventions and weight maintenance. The number of qualitative studies has also increased. Self-control and sleeve gastrectomy are also interesting research trends. A virtual model of weight achievement and intensive treatment employed during COVID-19 is also effective for weight reduction and maintenance [62].

#### 3.3.5. Research Cooperation

The country cooperation based on co-authorship is shown in Figure 5. Evidently, 22 countries have published 20 or more papers. The United Kingdom and Australia (*n* = 43), the United States and Canada (*n* = 28), the United States and the United Kingdom (*n* = 21), and the United States and Australia (*n* = 15) were the countries with the highest number of international publications. The most cited publications were from the United States, the United Kingdom, and Australia. The oldest publications were from the United States, Italy, Switzerland, and Sweden, and the youngest from Poland, Denmark, Saudi Arabia, and South Korea (Figure 5). Research cooperation on the motivation for weight loss between developed and less-developed countries is almost nonexistent.

## 4. Discussion

Over the past century, weight loss and motivation did not represent an interesting research topic. The year 2000 saw exponential growth in research productivity that has led to an increase in the number of publications. We believe that this research was primarily driven by the World Health Organization, with the exposure of the newly emerging health threat of obesity, which grew by 50% from 200 million to 300 million people in only five years [63].

Our study revealed that the research literature on motivations for weight loss is increasing, as is the scope of the research. Nevertheless, we must mention certain limitations. First, we performed the search in only one database; had we selected two or more, we might have harvested more research publications. On the other hand, Scopus (Elsevier, Amsterdam, The Netherlands) is the largest abstract and citation database of peer-reviewed literature. Additionally, content analysis is subjective, and other groups of researchers may reach slightly different conclusions. However, we employed a novel knowledge synthesis method through which we were able to analyze many relevant publications, thus minimizing publication selection bias. Thus, the research community is growing. This indicates that this research is gaining popularity and importance. However, we observed a regional concentration of research and its funding in developed countries. Cooperation between developed and less-developed countries is almost nonexistent. To improve this situation, a more intensive collaboration between low- and high-income countries must be established to support the integration of obesity research, interdisciplinary knowledge development, and evidence-based policymaking. In addition, collaborative research might contribute to a more comprehensive understanding of weight-loss problems and to building a shared base of data, innovations, and research paradigms. The qualitative part of the knowledge synthesis resulted in four themes and ten categories, indicating that the major problem in weight loss and management might be the maintenance of weight loss; this is relevant not only during the intervention, but also afterward when participants no longer have the supportive environment of the group and lose motivation to maintain their weight. Our synthesis revealed that weight regain might be a consequence of the fact that most patients cannot maintain positive lifestyle changes in general and transition back to unhealthy behavior due to psychological tension inflicted by conflicts between their existing habits and the healthy habits required to maintain weight loss. Consequently, the motivation for weight management and weight maintenance should be sustained after the intervention by adopting a healthy lifestyle, which includes knowledge of nutrition resulting in adaptive food restrictions; exercise, and physical and mental activity; a positive attitude toward change, acquiring a positive self-image and self-control; and the right kind of motivational activities. Another important conclusion of our study is that effective and efficient integration of diet, physical activity, and weight management services into primary care might reduce the burden of obesity. This can be achieved, for example, by integrating obesity management clinical pathways into health services [64], integrating community nursing and weight management services [65], or including novel lifestyle modification programs in primary care [66].

### Study Strengths and Limitations

The landscapes uncovered in this study present a multidimensionally faceted map of the weight loss and motivation problem, which can help the community solve theoretical and practical challenges. Obesity researchers and practitioners can use the study results to enhance their capacity of the area and can catalyze the further development of their knowledge. On the other hand, it can inform novice researchers, interested readers, research managers, or patients without specific knowledge and help them to develop a perspective on the most important dimensions of weight loss research. Finally, the landscapes can serve as a guide for further research and a starting point for more formal knowledge synthesis endeavors such as systematic reviews and meta-analyses.

## 5. Conclusions

The main outcome of our holistic study, in which we analyzed more than 2000 publications, is the identification of multidimensional facets and scientific landscapes of weight-loss research associated with motivation, which can help the weight and obesity research community in solving the theoretical and practical challenges and advancing knowledge development. The weight loss and motivation research landscapes consist of four themes: (1) weight loss and weight-loss maintenance through motivational interventions for lifestyle changes supported by smart ICT identified by the most popular author keywords—weight loss and physical activity; (2) motivational interviewing and health promotion; (3) maintaining a sustainable weight with a healthier lifestyle identified by author keywords—diet, diabetes, and lifestyle; and (4) weight maintenance and metabolic syndrome.

Researchers and practitioners in the field can utilize the study outcomes to improve their understanding of weight management and find prolific research partners and sponsoring agencies, while novice researchers, interested readers, and even patients with little or no expertise can gain insight into the most prominent weight management research dimensions. Finally, the study outcomes may guide further research and inform more traditional knowledge synthesis efforts.

## Figures and Tables

**Figure 1 healthcare-11-03086-f001:**
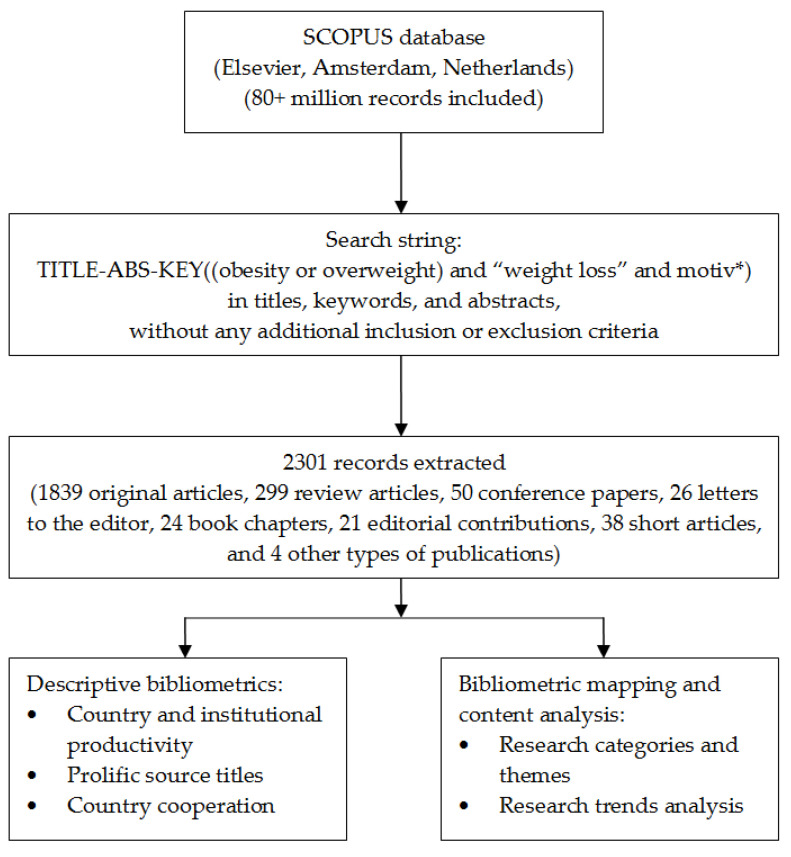
Search process flowchart.

**Figure 2 healthcare-11-03086-f002:**
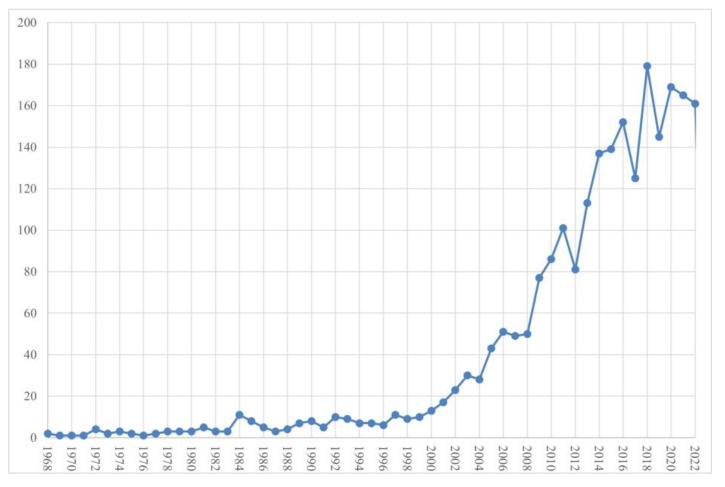
Dynamics of research literature on motivation in obesity and overweight.

**Figure 3 healthcare-11-03086-f003:**
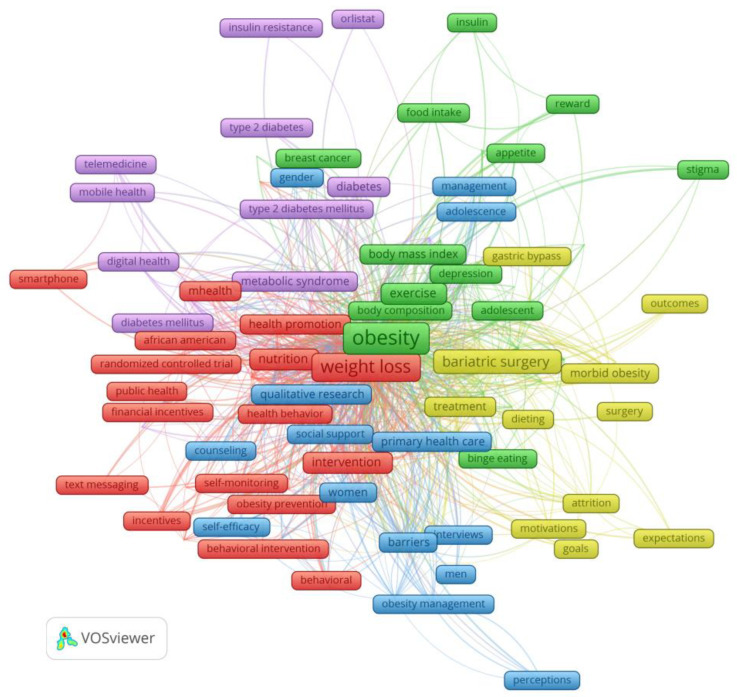
Author keyword map.

**Figure 4 healthcare-11-03086-f004:**
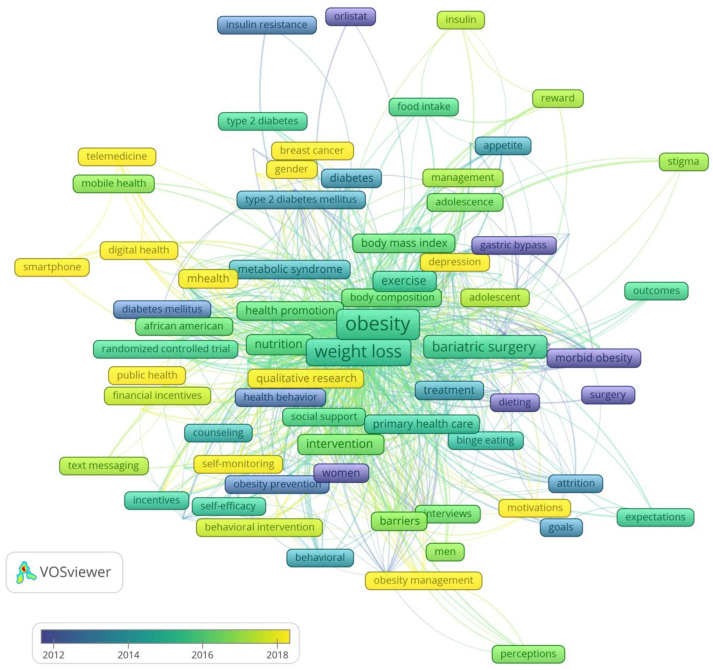
Trends in motivation in weight loss research.

**Figure 5 healthcare-11-03086-f005:**
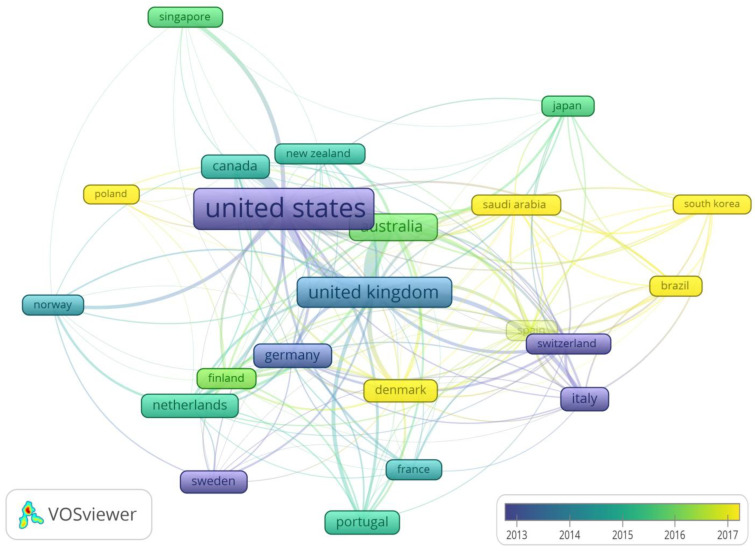
Country cooperation map based on co-authorships. The colors represent the average age of the publications, and the rectangle size is proportional to the number of citations.

**Table 1 healthcare-11-03086-t001:** Ten most productive countries.

Country	Number of Publications	Rank of Countries According to the Percentage of Obese People in 2022
United States of America	1070	21
United Kingdom	297	36
Australia	178	33
Canada	101	37
Germany	95	80
Netherlands	84	111
Italy	76	106
Spain	59	84
Denmark	52	121
France	50	80

**Table 2 healthcare-11-03086-t002:** Most prolific journals.

Journal	Number of Publications	Impact Factors (SJR—Scopus 2021)	H-Index	Quarter
*Obesity*	57	1.68	209	Q1
*Obesity Surgery*	54	0.96	143	Q1
*International Journal of Obesity*	41	1.38	234	Q1
*BMC Public Health*	31	1.16	159	Q1
*Contemporary Clinical Trials*	30	0.87	65	Q1
*Obesity Reviews*	29	2.54	172	Q1
*Appetite*	28	0.99	156	Q1
*Surgery for Obesity and Related Diseases*	28	1.11	93	Q1
*Obesity Research and Clinical Practice*	27	0.86	38	Q2
*Eating and Weight Disorders*	26	0.73	49	Q2

**Table 3 healthcare-11-03086-t003:** Representative author keywords, categories, and themes in research concerning motivation in weight loss.

Color	Representative Author Keywords (Codes)	Categories	Themes
Red	Weight loss (486), physical activity (157), motivational interviewing (67), adolescent (38), health promotion (30), lifestyle intervention (28), weight-loss maintenance (26), mHealth (24), obesity treatment (19)	Loss of weight through interventions in the form of motivational interviews, physical activity and changes in behavior and lifestyle supported by mHealth technologies; health promotion with motivational interviews to encourage physical activity; weight-loss maintenance with telemedicine and eHealth	Weight loss and weight-loss maintenance through motivational interventions; lifestyle changes supported by smart ICT
Violet	Diet (87), diabetes (49), lifestyle (34), weight maintenance (27), metabolic syndrome (22), weight control and reduction (22), prevention (21), digital health (14)	Nutrition and weight maintenance as prevention of metabolic diseases supported by digital health, weight maintenance as a lifestyle	Maintaining a sustainable weight with a healthier lifestyle
Blue	Overweight (157), weight management (92), primary care (61), qualitative research (36), barriers (27)	Weight management in overweight patients as part of primary care supported by qualitative research barriers to weight loss	Weight management on the level of primary healthcare
Yellow and Green	Obesity (860), bariatric surgery (140), motivation (118), body image (41), morbid obesity (30), treatment (30), self-regulation (26), internet (23), gastric bypass (18), dieting (18), weight regain (14)	Bariatric surgery and body self-image, weight regain after bariatric surgery, obesity and self-image	Bariatric surgery

## Data Availability

The original contributions presented in the study are included in the manuscript. The datasets generated during and/or analyzed during the current study are available from the corresponding author upon reasonable request.

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
