# Peer review of "Research Trends in Motivation and Weight Loss: A Bibliometric-Based Review"

_healthcare, 2023, doi:10.3390/healthcare11233086_

Round 1

Reviewer 1 Report

Comments and Suggestions for Authors

Dear Authors;

Thank you in advance for your scientific work.

Additional comments:

In relation to the questions:

1. The authors just used the following keywords in the searching: TITLE-ABS-KEY((obesity or overweight) and "weight loss" and motiv*). I think more synonyms and terms should be used to potentiate the knowledge and the manuscript objective.

2. Although the aim of the study was to do  a quantitative and qualitative bibliometric study that provides a holistic overview of research trends in motivation and weight loss, it is necessary to describe in a better way the flowchart and the included studies, mentioning i.e.: population age, population race, incomes, sport condition, among others. Here some examples: https://www.tandfonline.com/doi/abs/10.1080/09637486.2020.1868412; https://onlinelibrary.wiley.com/doi/10.1111/obr.12904

3. In addition, several manuscripts related to the study topic have been developed in the last few years. So, why are they not included?

4. English must be improved since some sentences seem to be cutted.

Sincerely,

Comments on the Quality of English Language

Dear Authors:

Moderate editing of English language required. Please, rewrite some sentences that seems to be cut.

Kind regards,

Author Response

Response to Reviewer #1 Comments

  1. The authors just used the following keywords in the searching: TITLE-ABS-KEY((obesity or overweight) and "weight loss" and motiv*). I think more synonyms and terms should be used to potentiate the knowledge and the manuscript objective.

Re:

The search was performed in a way to, as closely as possible, achieve the study objectives. In that manner, we formed the search string using the keywords focusing just on weight loss in obese or overweight people and motivation. We didn’t used synonyms because in that way irrelevant publications could be inserted in the corpus. However, we used wild character * and quotation marks. Thus, the sub search string "weight loss" retrieves all publications containing words weight loss, losing weight, weight loss and similar. We assessed the performance of the search string with the recall metric (the fraction of relevant instances among the relevant instances). In that manner, we formed the list of 30 fundamental publications related to the aim of the study and expected to be found in the corpus and 10 prolific authors. All 10 authors and 29 publications were successfully retrieved.

  1. Although the aim of the study was to do a quantitative and qualitative bibliometric study that provides a holistic overview of research trends in motivation and weight loss, it is necessary to describe in a better way the flowchart and the included studies, mentioning i.e.: population age, population race, incomes, sport condition, among others. Here some examples: https://www.tandfonline.com/doi/abs/10.1080/09637486.2020.1868412; https://onlinelibrary.wiley.com/doi/10.1111/obr.12904

Re:

Primarily papers included patients with various health problems as described between lines 193-198.

  1. In addition, several manuscripts related to the study topic have been developed in the last few years. So, why are they not included?

Re:

From the corpus consisting of more than 2300 papers we scoped and narratively reviewed the ones most in line with study aims and highly cited

  1. English must be improved since some sentences seem to be cutted.

Re:

Unfortunately, we could not find a native speaker, we would like to use MDPI Language Editing.

Reviewer 2 Report

Comments and Suggestions for Authors

Dear Authors,

Thank you for your manuscript. It is a nice work and I have some advice for you:

Your interesting work discusses the research trends in motivation and weight loss. However, I have not understood what is the population target of your review. Are they adolescents, adults or elderly? Please specify this point better.

Secondly, You just give very little consideration to physical activity. Only in the section “3.2.1.2. Health promotion with motivational interviews to encourage physical activity” line 199 you discuss physical activity and its power in fighting against obesity. However, this part should be spread because it is an important task when discussing weight loss management. What is the most suitable physical activity program? Are there any differences between resistance and endurance training? What is the best training frequency? I suggest you improve your article by inserting this missing part. To help you, I suggest these two interesting articles that can be useful for you to create a suitable section for this topic.

Fanelli E, Abate Daga F, Pappaccogli M, Eula E, Astarita A, Mingrone G, Fasano C, Magnino C, Schiavone D, Rabbone I, Gollin M, Rabbia F, Veglio F. A structured physical activity program in an adolescent population with overweight or obesity: a prospective interventional study. Appl Physiol Nutr Metab. 2022 Mar;47(3):253-260. doi: 10.1139/apnm-2021-0092. Epub 2021 Oct 27. PMID: 34706211.

Schranz, G. Tomkinson, N. Parletta, J. Petkov, and T. Olds, “Can resistance training change the strength, body composition and self-concept of overweight and obese adolescent males? A randomised controlled trial,” Br. J. Sports Med., vol. 48, no. 20, pp. 1482–1488, 2014

Thank you very much for your job.

Author Response

Response to Reviewer #2 Comments

  1. Your interesting work discusses the research trends in motivation and weight loss. However, I have not understood what is the population target of your review. Are they adolescents, adults or elderly? Please specify this point better.

Re:

The aim of our study was to present a holistic landscape of the research on weight loss motivation in obese people so we didn’t restrict the analysis and synthesis on any specific population. We extended the aim to point out this fact.

  1. Secondly, You just give very little consideration to physical activity. Only in the section “3.2.1.2. Health promotion with motivational interviews to encourage physical activity” line 199 you discuss physical activity and its power in fighting against obesity. However, this part should be spread because it is an important task when discussing weight loss management. What is the most suitable physical activity program? Are there any differences between resistance and endurance training? What is the best training frequency? I suggest you improve your article by inserting this missing part. To help you, I suggest these two interesting articles that can be useful for you to create a suitable section for this topic.

Fanelli E, Abate Daga F, Pappaccogli M, Eula E, Astarita A, Mingrone G, Fasano C, Magnino C, Schiavone D, Rabbone I, Gollin M, Rabbia F, Veglio F. A structured physical activity program in an adolescent population with overweight or obesity: a prospective interventional study. Appl Physiol Nutr Metab. 2022 Mar;47(3):253-260. doi: 10.1139/apnm-2021-0092. Epub 2021 Oct 27. PMID: 34706211.

Schranz, G. Tomkinson, N. Parletta, J. Petkov, and T. Olds, “Can resistance training change the strength, body composition and self-concept of overweight and obese adolescent males? A randomised controlled trial,” Br. J. Sports Med., vol. 48, no. 20, pp. 1482–1488, 2014

Re:

Chapter 3.2.1.2 was improved

Reviewer 3 Report

Comments and Suggestions for Authors

Dear authors,

I have reviewed your manuscript entitled: "Research Trends in Motivation and Weight Loss: A Bibliometric Based Review". The manuscript is very interesting and contributes to the development of the scientific literature in a novel way. I would like to take this opportunity to make several comments:

1) In the abstract, at the end of the introduction and at the beginning of the discussion, the research aims should be specifically mentioned. It is true that these aims can be interpreted, but making it more specific makes it easier to read. For example: "the research aim was: to study the Research Trends of the scientific literature on motivation and weight loss.

2) It would have been interesting to include more databases, such as PubMed. This would have broadened the analysis of the scientific literature and the review would have been of higher quality. You should bear this in mind for future research.

3) The introduction should include information on obesity, for example, definition, health effects, determinants, influence on different population groups, international recommendations, etc.

4) It would be useful to include what specific problems the authors have identified in order to carry out the research and what it contributes to the scientific field, not only to novel synthetics knowledge synthesis methodology. 

5) The size of figure 1 should be reduced, as it is excessive.

6) The structure of the manuscript can be much clearer. For example: 1. Introduction (information on obesity could be included here), 2. Method (explanation of the type of research and the steps followed), 3. Results (the sub-sections can be included as they already appear), 4. Discussion, 5. Conclusions.

7) The discussion section is not a repetition of the results already presented. Some explanation of the reasons for the results obtained should be included, even if they are assumptions based on some evidence. For example, is there any event or publication by an international body in the 2000s that has led to an increase in the number of publications? References should also be included to support the information provided. 

8) Expand the conclusions section, for example, by mentioning the most used keywords and the countries of residence of the authors of the manuscripts. 

I hope you will take my comments into account in order to improve the quality of the article. 

Best regards

Author Response

Response to Reviewer #3 Comments

  1. In the abstract, at the end of the introduction and at the beginning of the discussion, the research aims should be specifically mentioned. It is true that these aims can be interpreted, but making it more specific makes it easier to read. For example: "the research aim was: to study the Research Trends of the scientific literature on motivation and weight loss.

Re:

Abstract was improved.

  1. It would have been interesting to include more databases, such as PubMed. This would have broadened the analysis of the scientific literature, and the review would have been of higher quality. You should bear this in mind for future research.

Re:

Scopus is the largest bibliographic database of reviewed publications. It also includes Medline indexed publications and thus covers Pubmed.

  1. The introduction should include information on obesity, for example, definition, health effects, determinants, influence on different population groups, international recommendations, etc.

Re:

Introduction was expanded

  1. It would be useful to include what specific problems the authors have identified in order to carry out the research and what it contributes to the scientific field, not only to novel synthetics knowledge synthesis methodology. 

Re:

Study strengths and limitations were added to discussion section

  1. The size of figure 1 should be reduced, as it is excessive.

Re:

The size of Figure 1 was reduced

6 The structure of the manuscript can be much clearer. For example: 1. Introduction (information on obesity could be included here), 2. Method (explanation of the type of research and the steps followed), 3. Results (the sub-sections can be included as they already appear), 4. Discussion, 5. Conclusions.

Re:

Methods and Results were added as headings.

  1. The discussion section is not a repetition of the results already presented. Some explanation of the reasons for the results obtained should be included, even if they are assumptions based on some evidence. For example, is there any event or publication by an international body in the 2000s that has led to an increase in the number of publications? References should also be included to support the information provided.

Re:

Some explanation was added to discussion section.

8: Expand the conclusions section, for example, by mentioning the most used keywords and the countries of residence of the authors of the manuscripts. 

Re:

Conclusion was expanded:

Round 2

Reviewer 1 Report

Comments and Suggestions for Authors

Dear authors:

Thank you very much for providing us with the responses to our comments:

However, I still have several concerns about the previous suggestions:

  1. Although the authors provided us with the responses, it is still unclear the potential of the bibliometric review for scientific knowledge since the bias seems great if the suggestions are not addressed properly. Suggestions Nº1 and Nº3 are mandatory.
  2. No flowchart has been added. So, it is very difficult to understand the manuscript in this way.
  3. MDPI Language Editing must be considered to improve the manuscript's English.

Kind regards,

Comments on the Quality of English Language

Dear authors:

  1. MDPI Language Editing must be considered to improve the manuscript's English.

Kind regards,

Author Response

Response to Reviewer #1 Comments (round two)

Reviewer: However, I still have several concerns about the previous suggestions:

  1. Although the authors provided us with the responses, it is still unclear the potential of the bibliometric review for scientific knowledge since the bias seems great if the suggestions are not addressed properly. Suggestions Nº1 and Nº3 are mandatory.

Response: We included some description (lines 56-62). Inclusion of additional synonyms would result in many more extracted records; in additional/other keywords in connection to obesity and focus on motivation might be lost.

  1. No flowchart has been added. So, it is very difficult to understand the manuscript in this way.

Response: Flowchart was included.

  1. MDPI Language Editing must be considered to improve the manuscript's English.

Response: MDPI language editing services were used (English-Editing-Certificate-74360).

Reviewer 2 Report

Comments and Suggestions for Authors

Dear Authors,

thank you for your effort in improving your manuscript.

Actually, I beleave that it is much better than before and this can positively affect the impact of your research.  Considering that I am not english native speaker, I suggest you to verify grammar and typos with a native english speaker or well qualified english writer to ensure no errors or mistakes in the use of English.

Author Response

Response to Reviewer #2 Comments (round two)

Reviewer: Actually, I believe that it is much better than before and this can positively affect the impact of your research.  Considering that I am not english native speaker, I suggest you to verify grammar and typos with a native english speaker or well qualified english writer to ensure no errors or mistakes in the use of English.

Response: MDPI language editing services were used (English-Editing-Certificate-74360).

Reviewer 3 Report

Comments and Suggestions for Authors

Dear authors,

Thank you for taking into account my comments from the first revision to improve the quality of the manuscript. The modifications you have made are correct.

Kind regards

Author Response

Response to Reviewer #3 Comments (round two)

Reviewer: Thank you for taking into account my comments from the first revision to improve the quality of the manuscript. The modifications you have made are correct.

Response: Thank you very much.

MDPI language editing services were used (English-Editing-Certificate-74360).